# A Multi-Batch L-BFGS Method for Machine Learning

**Albert S. Berahas**
Northwestern University
Evanston, IL
albertberahas@u.northwestern.edu

**Jorge Nocedal**
Northwestern University
Evanston, IL
j-nocedal@northwestern.edu

**Martin Takáč**
Lehigh University
Bethlehem, PA
takac.mt@gmail.com

## Abstract

The question of how to parallelize the stochastic gradient descent (SGD) method has received much attention in the literature. In this paper, we focus instead on batch methods that use a sizeable fraction of the training set at each iteration to facilitate parallelism, and that employ second-order information. In order to improve the learning process, we follow a *multi-batch* approach in which the batch changes at each iteration. This can cause difficulties because L-BFGS employs gradient differences to update the Hessian approximations, and when these gradients are computed using different data points the process can be unstable. This paper shows how to perform stable quasi-Newton updating in the multi-batch setting, illustrates the behavior of the algorithm in a distributed computing platform, and studies its convergence properties for both the convex and nonconvex cases.

## 1   Introduction

It is common in machine learning to encounter optimization problems involving millions of parameters and very large datasets. To deal with the computational demands imposed by such applications, high performance implementations of stochastic gradient and batch quasi-Newton methods have been developed [1, 11, 9]. In this paper we study a batch approach based on the L-BFGS method [20] that strives to reach the right balance between efficient learning and productive parallelism.

In supervised learning, one seeks to minimize empirical risk,

$$F(w) := \frac{1}{n}\sum_{i=1}^{n} f(w; x^i, y^i) \stackrel{\text{def}}{=} \frac{1}{n}\sum_{i=1}^{n} f_i(w),$$

where $(x^i, y^i)_{i=1}^{n}$ denote the training examples and $f(\cdot; x, y) : \mathbb{R}^d \to \mathbb{R}$ is the composition of a prediction function (parametrized by $w$) and a loss function. The training problem consists of finding an optimal choice of the parameters $w \in \mathbb{R}^d$ with respect to $F$, i.e.,

$$\min_{w\in\mathbb{R}^d} F(w) = \frac{1}{n}\sum_{i=1}^{n} f_i(w). \tag{1.1}$$

At present, the preferred optimization method is the stochastic gradient descent (SGD) method [23, 5], and its variants [14, 24, 12], which are implemented either in an asynchronous manner (e.g. when

using a parameter server in a distributed setting) or following a synchronous mini-batch approach that exploits parallelism in the gradient evaluation [2, 22, 13]. A drawback of the asynchronous approach is that it cannot use large batches, as this would cause updates to become too dense and compromise the stability and scalability of the method [16, 22]. As a result, the algorithm spends more time in communication as compared to computation. On the other hand, using a synchronous mini-batch approach one can achieve a near-linear decrease in the number of SGD iterations as the mini-batch size is increased, up to a certain point after which the increase in computation is not offset by the faster convergence [26].

An alternative to SGD is a batch method, such as L-BFGS, which is able to reach high training accuracy and allows one to perform more computation per node, so as to achieve a better balance with communication costs [27]. Batch methods are, however, not as efficient learning algorithms as SGD in a sequential setting [6]. To benefit from the strength of both methods some high performance systems employ SGD at the start and later switch to a batch method [1].

**Multi-Batch Method.** In this paper, we follow a different approach consisting of a single method that selects a *sizeable* subset (batch) of the training data to compute a step, and changes this batch at each iteration to improve the learning abilities of the method. We call this a *multi-batch* approach to differentiate it from the mini-batch approach used in conjunction with SGD, which employs a very small subset of the training data. When using large batches it is natural to employ a quasi-Newton method, as incorporating second-order information imposes little computational overhead and improves the stability and speed of the method. We focus here on the L-BFGS method, which employs gradient information to update an estimate of the Hessian and computes a step in $O(d)$ flops, where $d$ is the number of variables. The multi-batch approach can, however, cause difficulties to L-BFGS because this method employs gradient differences to update Hessian approximations. When the gradients used in these differences are based on different data points, the updating procedure can be unstable. Similar difficulties arise in a parallel implementation of the standard L-BFGS method, if some of the computational nodes devoted to the evaluation of the function and gradient are unable to return results on time — as this again amounts to using different data points to evaluate the function and gradient at the beginning and the end of the iteration. The goal of this paper is to show that stable quasi-Newton updating can be achieved in both settings without incurring extra computational cost, or special synchronization. The key is to perform quasi-Newton updating based on the overlap between consecutive batches. The only restriction is that this overlap should not be too small, something that can be achieved in most situations.

**Contributions.** We describe a novel implementation of the batch L-BFGS method that is robust in the absence of sample consistency; i.e., when different samples are used to evaluate the objective function and its gradient at consecutive iterations. The numerical experiments show that the method proposed in this paper — which we call the *multi-batch* L-BFGS method — achieves a good balance between computation and communication costs. We also analyze the convergence properties of the new method (using a fixed step length strategy) on both convex and nonconvex problems.

## 2 The Multi-Batch Quasi-Newton Method

In a pure batch approach, one applies a gradient based method, such as L-BFGS [20], to the deterministic optimization problem (1.1). When the number $n$ of training examples is large, it is natural to parallelize the evaluation of $F$ and $\nabla F$ by assigning the computation of the component functions $f_i$ to different processors. If this is done on a distributed platform, it is possible for some of the computational nodes to be slower than the rest. In this case, the contribution of the slow (or unresponsive) computational nodes could be ignored given the stochastic nature of the objective function. This leads, however, to an inconsistency in the objective function and gradient at the beginning and at the end of the iteration, which can be detrimental to quasi-Newton methods. Thus, we seek to find a *fault-tolerant* variant of the batch L-BFGS method that is capable of dealing with slow or unresponsive computational nodes.

A similar challenge arises in a *multi-batch* implementation of the L-BFGS method in which the entire training set $T = \left\{ (x^i, y^i)_{i=1}^n \right\}$ is not employed at every iteration, but rather, a subset of the data is used to compute the gradient. Specifically, we consider a method in which the dataset is randomly divided into a number of batches — say 10, 50, or 100 — and the minimization is performed with respect to a different batch at every iteration. At the $k$-th iteration, the algorithm chooses a batch

$S_k \subset \{1, \ldots, n\}$, computes

$$F^{S_k}(w_k) = \frac{1}{|S_k|} \sum_{i \in S_k} f_i(w_k), \qquad \nabla F^{S_k}(w_k) = g_k^{S_k} = \frac{1}{|S_k|} \sum_{i \in S_k} \nabla f_i(w_k), \qquad (2.2)$$

and takes a step along the direction $-H_k g_k^{S_k}$, where $H_k$ is an approximation to $\nabla^2 F(w_k)^{-1}$. Allowing the sample $S_k$ to change freely at every iteration gives this approach flexibility of implementation and is beneficial to the learning process, as we show in Section 4. (We refer to $S_k$ as the sample of training points, even though $S_k$ only indexes those points.)

The case of unresponsive computational nodes and the multi-batch method are similar. The main difference is that node failures create unpredictable changes to the samples $S_k$, whereas a multi-batch method has control over sample generation. In either case, the algorithm employs a stochastic approximation to the gradient and can no longer be considered deterministic. We must, however, distinguish our setting from that of the classical SGD method, which employs small mini-batches and noisy gradient approximations. Our algorithm operates with much larger batches so that distributing the function evaluation is beneficial and the compute time of $g_k^{S_k}$ is not overwhelmed by communication costs. This gives rise to gradients with relatively small variance and justifies the use of a second-order method such as L-BFGS.

**Robust Quasi-Newton Updating.** The difficulties created by the use of a different sample $S_k$ at each iteration can be circumvented if consecutive samples $S_k$ and $S_{k+1}$ overlap, so that $O_k = S_k \cap S_{k+1} \neq \emptyset$. One can then perform stable quasi-Newton updating by computing gradient differences based on this overlap, i.e., by defining

$$y_{k+1} = g_{k+1}^{O_k} - g_k^{O_k}, \qquad s_{k+1} = w_{k+1} - w_k, \qquad (2.3)$$

in the notation given in (2.2). The correction pair $(y_k, s_k)$ can then be used in the BFGS update. When the overlap set $O_k$ is not too small, $y_k$ is a useful approximation of the curvature of the objective function $F$ along the most recent displacement, and will lead to a productive quasi-Newton step. This observation is based on an important property of Newton-like methods, namely that there is much more freedom in choosing a Hessian approximation than in computing the gradient [7, 3]. Thus, a smaller sample $O_k$ can be employed for updating the inverse Hessian approximation $H_k$ than for computing the batch gradient $g_k^{S_k}$ in the search direction $-H_k g_k^{S_k}$. In summary, by ensuring that unresponsive nodes do not constitute the vast majority of all working nodes in a fault-tolerant parallel implementation, or by exerting a small degree of control over the creation of the samples $S_k$ in the multi-batch method, one can design a robust method that naturally builds upon the fundamental properties of BFGS updating.

We should mention in passing that a commonly used strategy for ensuring stability of quasi-Newton updating in machine learning is to enforce gradient consistency [25], i.e., to use the same sample $S_k$ to compute gradient evaluations at the beginning and the end of the iteration. Another popular remedy is to use the same batch $S_k$ for multiple iterations [19], alleviating the gradient inconsistency problem at the price of slower convergence. In this paper, we assume that achieving such *sample consistency is not possible* (in the fault-tolerant case) or *desirable* (in a multi-batch framework), and wish to design a new variant of L-BFGS that imposes minimal restrictions in the sample changes.

## 2.1 Specification of the Method

At the $k$-th iteration, the multi-batch BFGS algorithm chooses a set $S_k \subset \{1, \ldots, n\}$ and computes a new iterate

$$w_{k+1} = w_k - \alpha_k H_k g_k^{S_k}, \qquad (2.4)$$

where $\alpha_k$ is the step length, $g_k^{S_k}$ is the batch gradient (2.2) and $H_k$ is the inverse BFGS Hessian matrix approximation that is updated at every iteration by means of the formula

$$H_{k+1} = V_k^T H_k V_k + \rho_k s_k s_k^T, \qquad \rho_k = \frac{1}{y_k^T s_k}, \qquad V_k = I - \rho_k y_k s_k^T.$$

To compute the correction vectors $(s_k, y_k)$, we determine the overlap set $O_k = S_k \cap S_{k+1}$ consisting of the samples that are common at the $k$-th and $k+1$-st iterations. We define

$$F^{O_k}(w_k) = \frac{1}{|O_k|} \sum_{i \in O_k} f_i(w_k), \qquad \nabla F^{O_k}(w_k) = g_k^{O_k} = \frac{1}{|O_k|} \sum_{i \in O_k} \nabla f_i(w_k),$$

and compute the correction vectors as in (2.3). In this paper we assume that $\alpha_k$ is constant.

In the limited memory version, the matrix $H_k$ is defined at each iteration as the result of applying $m$ BFGS updates to a multiple of the identity matrix, using a set of $m$ correction pairs $\{s_i, y_i\}$ kept in storage. The memory parameter $m$ is typically in the range 2 to 20. When computing the matrix-vector product in (2.4) it is not necessary to form that matrix $H_k$ since one can obtain this product via the two-loop recursion [20], using the $m$ most recent correction pairs $\{s_i, y_i\}$. After the step has been computed, the oldest pair $(s_j, y_j)$ is discarded and the new curvature pair is stored.

A pseudo-code of the proposed method is given below, and depends on several parameters. The parameter $r$ denotes the fraction of samples in the dataset used to define the gradient, i.e., $r = \frac{|S|}{n}$. The parameter $o$ denotes the length of overlap between consecutive samples, and is defined as a fraction of the number of samples in a given batch $S$, i.e., $o = \frac{|O|}{|S|}$.

---

**Algorithm 1** Multi-Batch L-BFGS

---

**Input:** $w_0$ (initial iterate), $T = \{(x^i, y^i), \text{ for } i = 1, \ldots, n\}$ (training set), $m$ (memory parameter), $r$ (batch, fraction of $n$), $o$ (overlap, fraction of batch), $k \leftarrow 0$ (iteration counter).

1: Create initial batch $S_0$                                                  ▷ As shown in Firgure 1
2: **for** $k = 0, 1, 2, \ldots$ **do**
3:      Calculate the search direction $p_k = -H_k g_k^{S_k}$              ▷ Using L-BFGS formula
4:      Choose the step length $\alpha_k > 0$
5:      Compute $w_{k+1} = w_k + \alpha_k p_k$
6:      Create the next batch $S_{k+1}$
7:      Compute the curvature pairs $s_{k+1} = w_{k+1} - w_k$ and $y_{k+1} = g_{k+1}^{O_k} - g_k^{O_k}$
8:      Replace the oldest pair $(s_i, y_i)$ by $s_{k+1}, y_{k+1}$
9: **end for**

---

## 2.2 Sample Generation

We now discuss how the sample $S_{k+1}$ is created at each iteration (Line 8 in Algorithm 1).

**Distributed Computing with Faults.** Consider a distributed implementation in which slave nodes read the current iterate $w_k$ from the master node, compute a local gradient on a subset of the dataset, and send it back to the master node for aggregation in the calculation (2.2). Given a time (computational) budget, it is possible for some nodes to fail to return a result. The schematic in Figure 1a illustrates the gradient calculation across two iterations, $k$ and $k+1$, in the presence of faults. Here $\mathcal{B}_i$, $i = 1, \ldots, B$ denote the batches of data that each slave node $i$ receives (where $T = \cup_i \mathcal{B}_i$), and $\tilde{\nabla} f(w)$ is the gradient calculation using all nodes that responded within the preallocated time.

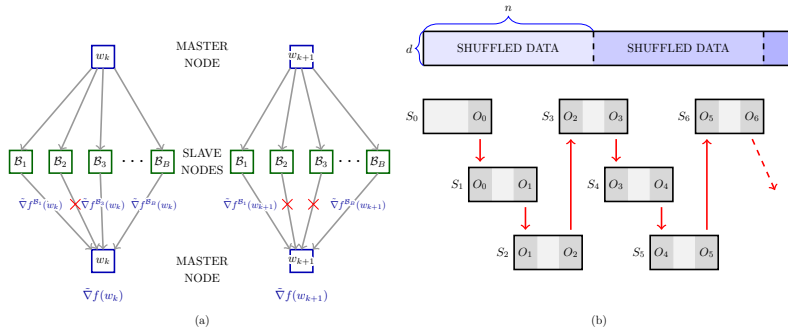

Figure 1: Sample and Overlap formation.

Let $\mathcal{J}_k \subset \{1, 2, \ldots, B\}$ and $\mathcal{J}_{k+1} \subset \{1, 2, \ldots, B\}$ be the set of indices of all nodes that returned a gradient at the $k$-th and $k+1$-st iterations, respectively. Using this notation $S_k = \cup_{j \in \mathcal{J}_k} \mathcal{B}_j$ and $S_{k+1} = \cup_{j \in \mathcal{J}_{k+1}} \mathcal{B}_j$, and we define $O_k = \cup_{j \in \mathcal{J}_k \cap \mathcal{J}_{k+1}} \mathcal{B}_j$. The simplest implementation in this setting preallocates the data on each compute node, requiring minimal data communication, i.e., only

one data transfer. In this case the samples $S_k$ will be independent if node failures occur randomly. On the other hand, if the same set of nodes fail, then sample creation will be biased, which is harmful both in theory and practice. One way to ensure independent sampling is to shuffle and redistribute the data to all nodes after a certain number of iterations.

**Multi-batch Sampling.** We propose two strategies for the *multi-batch* setting.

Figure 1b illustrates the sample creation process in the first strategy. The dataset is shuffled and batches are generated by collecting subsets of the training set, in order. Every set (except $S_0$) is of the form $S_k = \{O_{k-1}, N_k, O_k\}$, where $O_{k-1}$ and $O_k$ are the overlapping samples with batches $S_{k-1}$ and $S_{k+1}$ respectively, and $N_k$ are the samples that are unique to batch $S_k$. After each pass through the dataset, the samples are reshuffled, and the procedure described above is repeated. In our implementation samples are drawn without replacement, guaranteeing that after every pass (epoch) all samples are used. This strategy has the advantage that it requires no extra computation in the evaluation of $g_k^{O_k}$ and $g_{k+1}^{O_k}$, but the samples $\{S_k\}$ are not independent.

The second sampling strategy is simpler and requires less control. At every iteration $k$, a batch $S_k$ is created by randomly selecting $|S_k|$ elements from $\{1, \ldots n\}$. The overlapping set $O_k$ is then formed by randomly selecting $|O_k|$ elements from $S_k$ (subsampling). This strategy is slightly more expensive since $g_{k+1}^{O_k}$ requires extra computation, but if the overlap is small this cost is not significant.

## 3 Convergence Analysis

In this section, we analyze the convergence properties of the multi-batch L-BFGS method (Algorithm 1) when applied to the minimization of *strongly convex* and *nonconvex* objective functions, using a fixed step length strategy. We assume that the goal is to minimize the empirical risk $F$ given in (1.1), but note that a similar analysis could be used to study the minimization of the expected risk.

### 3.1 Strongly Convex case

Due to the stochastic nature of the multi-batch approach, every iteration of Algorithm 1 employs a gradient that contains errors that do not converge to zero. Therefore, by using a fixed step length strategy one cannot establish convergence to the optimal solution $w^\star$, but only convergence to a neighborhood of $w^\star$ [18]. Nevertheless, this result is of interest as it reflects the common practice of using a fixed step length and decreasing it only if the desired testing error has not been achieved. It also illustrates the tradeoffs that arise between the size of the batch and the step length.

In our analysis, we make the following assumptions about the objective function and the algorithm.

**Assumptions A.**

1. *$F$ is twice continuously differentiable.*
2. *There exist positive constants $\hat{\lambda}$ and $\hat{\Lambda}$ such that $\hat{\lambda}I \preceq \nabla^2 F^O(w) \preceq \hat{\Lambda}I$ for all $w \in \mathbb{R}^d$ and all sets $O \subset \{1, 2, \ldots, n\}$.*
3. *There is a constant $\gamma$ such that $\mathbb{E}_S\left[\|\nabla F^S(w)\|\right]^2 \leq \gamma^2$ for all $w \in \mathbb{R}^d$ and all sets $S \subset \{1, 2, \ldots, n\}$.*
4. *The samples $S$ are drawn independently and $\nabla F^S(w)$ is an unbiased estimator of the true gradient $\nabla F(w)$ for all $w \in \mathbb{R}^d$, i.e., $\mathbb{E}_S[\nabla F^S(w)] = \nabla F(w)$.*

Note that Assumption $A.2$ implies that the entire Hessian $\nabla^2 F(w)$ also satisfies

$$\lambda I \preceq \nabla^2 F(w) \preceq \Lambda I, \quad \forall w \in \mathbb{R}^d,$$

for some constants $\lambda, \Lambda > 0$. Assuming that every sub-sampled function $F^O(w)$ is strongly convex is not unreasonable as a regularization term is commonly added in practice when that is not the case.

We begin by showing that the inverse Hessian approximations $H_k$ generated by the multi-batch L-BFGS method have eigenvalues that are uniformly bounded above and away from zero. The proof technique used is an adaptation of that in [8].

**Lemma 3.1.** *If Assumptions A.1-A.2 above hold, there exist constants $0 < \mu_1 \leq \mu_2$ such that the Hessian approximations $\{H_k\}$ generated by Algorithm 1 satisfy*

$$\mu_1 I \preceq H_k \preceq \mu_2 I, \qquad for\ k = 0, 1, 2, \ldots$$

Utilizing Lemma 3.1, we show that the multi-batch L-BFGS method with a constant step length converges to a neighborhood of the optimal solution.

**Theorem 3.2.** *Suppose that Assumptions A.1-A.4 hold and let $F^\star = F(w^\star)$, where $w^\star$ is the minimizer of $F$. Let $\{w_k\}$ be the iterates generated by Algorithm 1 with $\alpha_k = \alpha \in (0, \frac{1}{2\mu_1\lambda})$, starting from $w_0$. Then for all $k \geq 0$,*

$$\mathbb{E}[F(w_k) - F^\star] \leq (1 - 2\alpha\mu_1\lambda)^k [F(w_0) - F^\star] + [1 - (1 - \alpha\mu_1\lambda)^k] \frac{\alpha\mu_2^2\gamma^2\Lambda}{4\mu_1\lambda}$$

$$\xrightarrow{k\to\infty} \frac{\alpha\mu_2^2\gamma^2\Lambda}{4\mu_1\lambda}.$$

The bound provided by this theorem has two components: (i) a term decaying linearly to zero, and (ii) a term identifying the neighborhood of convergence. Note that a larger step length yields a more favorable constant in the linearly decaying term, at the cost of an increase in the size of the neighborhood of convergence. We will consider again these tradeoffs in Section 4, where we also note that larger batches increase the opportunities for parallelism and improve the limiting accuracy in the solution, but slow down the learning abilities of the algorithm.

One can establish convergence of the multi-batch L-BFGS method to the optimal solution $w^\star$ by employing a sequence of step lengths $\{\alpha_k\}$ that converge to zero according to the schedule proposed by Robbins and Monro [23]. However, that provides only a sublinear rate of convergence, which is of little interest in our context where large batches are employed and some type of linear convergence is expected. In this light, Theorem 3.2 is more relevant to practice.

## 3.2 Nonconvex case

The BFGS method is known to fail on noconvex problems [17, 10]. Even for L-BFGS, which makes only a finite number of updates at each iteration, one cannot guarantee that the Hessian approximations have eigenvalues that are uniformly bounded above and away from zero. To establish convergence of the BFGS method in the nonconvex case *cautious* updating procedures have been proposed [15]. Here we employ a cautious strategy that is well suited to our particular algorithm; we skip the update, i.e., set $H_{k+1} = H_k$, if the curvature condition

$$y_k^T s_k \geq \epsilon\|s_k\|^2 \tag{3.5}$$

is not satisfied, where $\epsilon > 0$ is a predetermined constant. Using said mechanism we show that the eigenvalues of the Hessian matrix approximations generated by the multi-batch L-BFGS method are bounded above and away from zero (Lemma 3.3). The analysis presented in this section is based on the following assumptions.

**Assumptions B.**

1. *$F$ is twice continuously differentiable.*
2. *The gradients of $F$ are $\Lambda$-Lipschitz continuous, and the gradients of $F^O$ are $\Lambda_O$-Lipschitz continuous for all $w \in \mathbb{R}^d$ and all sets $O \subset \{1, 2, \ldots, n\}$.*
3. *The function $F(w)$ is bounded below by a scalar $\widehat{F}$.*
4. *There exist constants $\gamma \geq 0$ and $\eta > 0$ such that $\mathbb{E}_S \left[\|\nabla F^S(w)\|\right]^2 \leq \gamma^2 + \eta\|\nabla F(w)\|^2$ for all $w \in \mathbb{R}^d$ and all sets $S \subset \{1, 2, \ldots, n\}$.*
5. *The samples $S$ are drawn independently and $\nabla F^S(w)$ is an unbiased estimator of the true gradient $\nabla F(w)$ for all $w \in \mathbb{R}^d$, i.e., $\mathbb{E}[\nabla F^S(w)] = \nabla F(w)$.*

**Lemma 3.3.** *Suppose that Assumptions B.1-B.2 hold and let $\epsilon > 0$ be given. Let $\{H_k\}$ be the Hessian approximations generated by Algorithm 1, with the modification that $H_{k+1} = H_k$ whenever (3.5) is not satisfied. Then, there exist constants $0 < \mu_1 \leq \mu_2$ such that*

$$\mu_1 I \preceq H_k \preceq \mu_2 I, \qquad \text{for } k = 0, 1, 2, \ldots$$

We can now follow the analysis in [4, Chapter 4] to establish the following result about the behavior of the gradient norm for the multi-batch L-BFGS method with a cautious update strategy.

**Theorem 3.4.** *Suppose that Assumptions B.1-B.5 above hold, and let $\epsilon > 0$ be given. Let $\{w_k\}$ be the iterates generated by Algorithm 1, with $\alpha_k = \alpha \in (0, \frac{\mu_1}{\mu_2^2\eta\Lambda})$, starting from $w_0$, and with the*

*modification that $H_{k+1} = H_k$ whenever (3.5) is not satisfied. Then,*

$$\mathbb{E}\Big[\frac{1}{L}\sum_{k=0}^{L-1}\|\nabla F(w_k)\|^2\Big] \leq \frac{\alpha\mu_2^2\gamma^2\Lambda}{\mu_1} + \frac{2[F(w_0)-\widehat{F}]}{\alpha\mu_1 L}$$

$$\xrightarrow{L\to\infty} \frac{\alpha\mu_2^2\gamma^2\Lambda}{\mu_1}.$$

This result bounds the average norm of the gradient of $F$ after the first $L-1$ iterations, and shows that the iterates spend increasingly more time in regions where the objective function has a small gradient.

## 4    Numerical Results

In this Section, we present numerical results that evaluate the proposed robust multi-batch L-BFGS scheme (Algorithm 1) on logistic regression problems. Figure 2 shows the performance on the webspam dataset[1], where we compare it against three methods: (i) multi-batch L-BFGS without enforcing sample consistency (L-BFGS), where gradient differences are computed using different samples, i.e., $y_k = g_{k+1}^{S_{k+1}} - g_k^{S_k}$; (ii) multi-batch gradient descent (Gradient Descent), which is obtained by setting $H_k = I$ in Algorithm 1; and, (iii) serial SGD, where at every iteration one sample is used to compute the gradient. We run each method with 10 different random seeds, and, where applicable, report results for different batch ($r$) and overlap ($o$) sizes. The proposed method is more stable than the standard L-BFGS method; this is especially noticeable when $r$ is small. On the other hand, serial SGD achieves similar accuracy as the robust L-BFGS method and at a similar rate (e.g., $r = 1\%$), at the cost of $n$ communications per epochs versus $\frac{1}{r(1-o)}$ communications per epoch. Figure 2 also indicates that the robust L-BFGS method is not too sensitive to the size of overlap. Similar behavior was observed on other datasets, in regimes where $r \cdot o$ was not too small. We mention in passing that the L-BFGS step was computed using the a vector-free implementation proposed in [9].

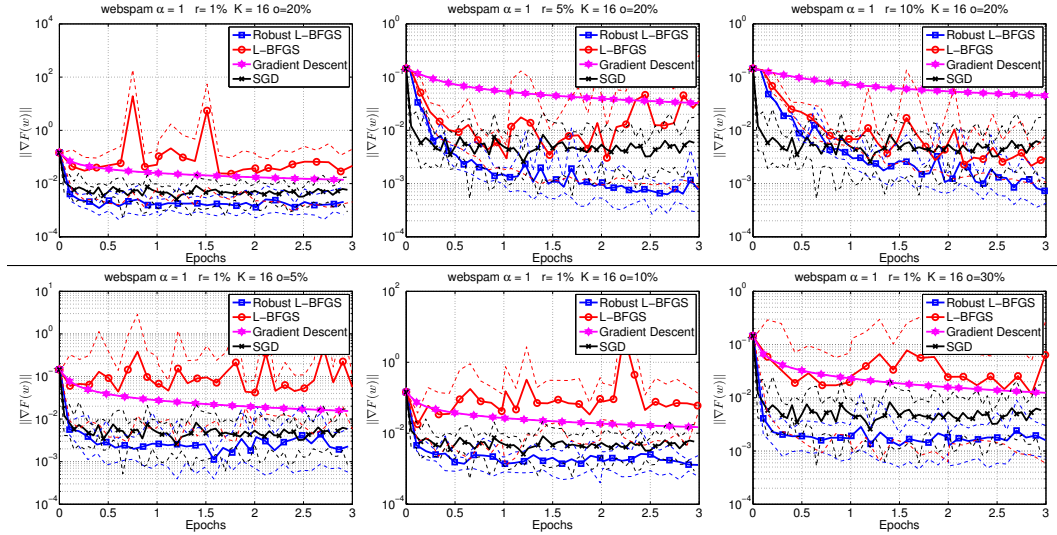

Figure 2: **webspam dataset**. Comparison of Robust L-BFGS, L-BFGS (multi-batch L-BFGS without enforcing sample consistency), Gradient Descent (multi-batch Gradient method) and SGD for various batch ($r$) and overlap ($o$) sizes. Solid lines show average performance, and dashed lines show worst and best performance, over 10 runs (per algorithm). $K = 16$ MPI processes.

We also explore the performance of the robust multi-batch L-BFGS method in the presence of node failures (faults), and compare it to the multi-batch variant that does not enforce sample consistency (L-BFGS). Figure 3 illustrates the performance of the methods on the webspam dataset, for various

probabilities of node failures $p \in \{0.1, 0.3, 0.5\}$, and suggests that the robust L-BFGS variant is more stable.

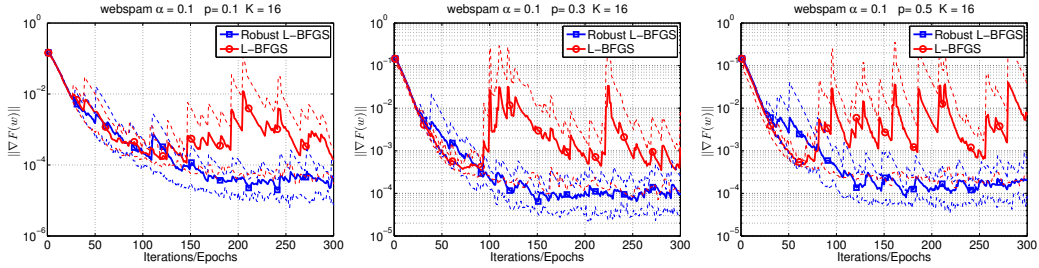

Figure 3: **webspam dataset**. Comparison of Robust L-BFGS and L-BFGS (multi-batch L-BFGS without enforcing sample consistency), for various node failure probabilities $p$. Solid lines show average performance, and dashed lines show worst and best performance, over 10 runs (per algorithm). $K = 16$ MPI processes.

Lastly, we study the strong and weak scaling properties of the robust L-BFGS method on artificial data (Figure 4). We measure the time needed to compute a gradient (Gradient) and the associated communication (Gradient+C), as well as, the time needed to compute the L-BFGS direction (L-BFGS) and the associated communication (L-BFGS+C), for various batch sizes ($r$). The figure on the left shows strong scaling of multi-batch LBFGS on a $d = 10^4$ dimensional problem with $n = 10^7$ samples. The size of input data is 24GB, and we vary the number of MPI processes, $K \in \{1, 2, \ldots, 128\}$. The time it takes to compute the gradient decreases with $K$, however, for small values of $r$, the communication time exceeds the compute time. The figure on the right shows weak scaling on a problem of similar size, but with varying sparsity. Each sample has $10 \cdot K$ non-zero elements, thus for any $K$ the size of local problem is roughly 1.5GB (for $K = 128$ size of data 192GB). We observe almost constant time for the gradient computation while the cost of computing the L-BFGS direction decreases with $K$; however, if communication is considered, the overall time needed to compute the L-BFGS direction increases slightly.

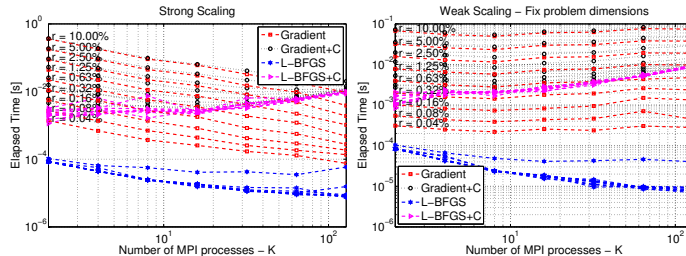

Figure 4: Strong and weak scaling of multi-batch L-BFGS method.

## 5 Conclusion

This paper describes a novel variant of the L-BFGS method that is robust and efficient in two settings. The first occurs in the presence of node failures in a distributed computing implementation; the second arises when one wishes to employ a different batch at each iteration in order to accelerate learning. The proposed method avoids the pitfalls of using inconsistent gradient differences by performing quasi-Newton updating based on the overlap between consecutive samples. Numerical results show that the method is efficient in practice, and a convergence analysis illustrates its theoretical properties.

**Acknowledgements**

The first two authors were supported by the Office of Naval Research award N000141410313, the Department of Energy grant DE-FG02-87ER25047 and the National Science Foundation grant DMS-1620022. Martin Takáč was supported by National Science Foundation grant CCF-1618717.

## Footnotes

[1]LIBSVM: `https://www.csie.ntu.edu.tw/~cjlin/libsvmtools/datasets/binary.html`.

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
