[Supplementary Material]

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

# A   Proofs and Technical Results

## A.1   Assumptions

We first restate the Assumptions that we use in the Convergence Analysis section (Section 3). Assumption $A$ and $B$ are used in the strongly convex and nonconvex cases, respectively.

### Assumptions A

1. $F$ is twice continuously differentiable.
2. There exist positive constants $\hat{\lambda}$ and $\hat{\Lambda}$ such that

$$\hat{\lambda} I \preceq \nabla^2 F^O(w) \preceq \hat{\Lambda} I, \tag{A.6}$$

   for all $w \in \mathbb{R}^d$ and all sets $O \subset \{1, 2, \ldots, n\}$.
3. There is a constant $\gamma$ such that

$$\mathbb{E}_S \left[ \|\nabla F^S(w)\| \right]^2 \leq \gamma^2, \tag{A.7}$$

   for all $w \in \mathbb{R}^d$ and all batches $S \subset \{1, 2, \ldots, n\}$.
4. The samples $S$ are drawn independently and $\nabla F^S(w)$ is an unbiased estimator of the true gradient $\nabla F(w)$ for all $w \in \mathbb{R}^d$, i.e.,

$$\mathbb{E} \left[ \nabla F^S(w) \right] = \nabla F(w). \tag{A.8}$$

Note that Assumption $A.2$ implies that the entire Hessian $\nabla^2 F(w)$ also satisfies

$$\lambda I \preceq \nabla^2 F(w) \preceq \Lambda I, \forall w \in \mathbb{R}^d, \tag{A.9}$$

for some constants $\lambda, \Lambda > 0$.

### Assumptions B

1. $F$ is twice continuously differentiable.
2. The gradients of $F$ are $\Lambda$-Lipschitz continuous and the gradients of $F^O$ are $\Lambda_O$-Lipschitz continuous for all $w \in \mathbb{R}^d$ and all sets $O \subset \{1, 2, \ldots, n\}$.
3. The function $F(w)$ is bounded below by a scalar $\widehat{F}$.
4. There exist constants $\gamma \geq 0$ and $\eta > 0$ such that

$$\mathbb{E}_S \left[ \|\nabla F^S(w)\| \right]^2 \leq \gamma^2 + \eta \|\nabla F(w)\|^2, \tag{A.10}$$

   for all $w \in \mathbb{R}^d$ and all batches $S \subset \{1, 2, \ldots, n\}$.
5. The samples $S$ are drawn independently and $\nabla F^S(w)$ is an unbiased estimator of the true gradient $\nabla F(w)$ for all $w \in \mathbb{R}^d$, i.e.,

$$\mathbb{E} \left[ \nabla F^S(w) \right] = \nabla F(w). \tag{A.11}$$

## A.2   Proof of Lemma 3.1 (Strongly Convex Case)

The following Lemma shows that the eigenvalues of the matrices generated by the multi-batch L-BFGS method are bounded above and away from zero if $F$ is strongly convex.

**Lemma 3.1.** *If Assumptions A.1-A.2 above hold, there exist constants $0 < \mu_1 \leq \mu_2$ such that the Hessian approximations $\{H_k\}$ generated by the multi-batch L-BFGS method (Algorithm 1) satisfy*

$$\mu_1 I \preceq H_k \preceq \mu_2 I, \qquad \text{for } k = 0, 1, 2, \ldots$$

*Proof.* Instead of analyzing the inverse Hessian approximation $H_k$, we study the direct Hessian approximation $B_k = H_k^{-1}$. In this case, the limited memory quasi-Newton updating formula is given as follows

1. Set $B_k^{(0)} = \frac{y_k^T y_k}{s_k^T y_k} I$ and $\tilde{m} = \min\{k, m\}$; where $m$ is the memory in L-BFGS.

2. For $i = 0, ..., \tilde{m} - 1$ set $j = k - \tilde{m} + 1 + i$ and compute

$$B_k^{(i+1)} = B_k^{(i)} - \frac{B_k^{(i)} s_j s_j^T B_k^{(i)}}{s_j^T B_k^{(i)} s_j} + \frac{y_j y_j^T}{y_j^T s_j}.$$

3. Set $B_{k+1} = B_k^{(\tilde{m})}$.

The curvature pairs $s_k$ and $y_k$ are updated via the following formulae

$$y_{k+1} = g_{k+1}^{O_k} - g_k^{O_k}, \qquad s_k = w_{k+1} - w_k. \tag{A.12}$$

A consequence of Assumption $A.2$ is that the eigenvalues of any sub-sampled Hessian ($|O|$ samples) are bounded above and away from zero. Utilizing this fact, the convexity of component functions and the definitions (A.12), we have

$$y_k^T s_k \geq \frac{1}{\hat{\Lambda}} \|y_k\|^2 \quad \Rightarrow \quad \frac{\|y_k\|^2}{y_k^T s_k} \leq \hat{\Lambda}. \tag{A.13}$$

On the other hand, strong convexity of the sub-sampled functions, the consequence of Assumption $A.2$ and definitions (A.12), provide a lower bound,

$$y_k^T s_k \leq \frac{1}{\hat{\lambda}} \|y_k\|^2 \quad \Rightarrow \quad \frac{\|y_k\|^2}{y_k^T s_k} \geq \hat{\lambda}. \tag{A.14}$$

Combining the upper and lower bounds (A.13) and (A.14)

$$\hat{\lambda} \leq \frac{\|y_k\|^2}{y_k^T s_k} \leq \hat{\Lambda}. \tag{A.15}$$

The above proves that the eigenvalues of the matrices $B_k^{(0)} = \frac{y_k^T y_k}{s_k^T y_k} I$ at the start of the L-BFGS update cycles are bounded above and away from zero, for all $k$. We now use a Trace-Determinant argument to show that the eigenvalues of $B_k$ are bounded above and away from zero.

Let $Tr(B)$ and $\det(B)$ denote the trace and determinant of matrix $B$, respectively, and set $j_i = k - \tilde{m} + i$. The trace of the matrix $B_{k+1}$ can be expressed as,

$$Tr(B_{k+1}) = Tr(B_k^{(0)}) - Tr \sum_{i=1}^{\tilde{m}} \left( \frac{B_k^{(i)} s_{j_i} s_{j_i}^T B_k^{(i)}}{s_{j_i}^T B_k^{(i)} s_{j_i}} \right) + Tr \sum_{i=1}^{\tilde{m}} \frac{y_{j_i} y_{j_i}^T}{y_{j_i}^T s_{j_i}}$$

$$\leq Tr(B_k^{(0)}) + \sum_{i=1}^{\tilde{m}} \frac{\|y_{j_i}\|^2}{y_{j_i}^T s_{j_i}}$$

$$\leq Tr(B_k^{(0)}) + \tilde{m}\hat{\Lambda}$$

$$\leq C_1, \tag{A.16}$$

for some positive constant $C_1$, where the inequalities above are due to (A.15), and the fact that the eigenvalues of the initial L-BFGS matrix $B_k^{(0)}$ are bounded above and away from zero.

Using a result due to Powell [21], the determinant of the matrix $B_{k+1}$ generated by the multi-batch L-BFGS method can be expressed as,

$$\det(B_{k+1}) = \det(B_k^{(0)}) \prod_{i=1}^{\tilde{m}} \frac{y_{j_i}^T s_{j_i}}{s_{j_i}^T B_k^{(i-1)} s_{j_i}}$$

$$= \det(B_k^{(0)}) \prod_{i=1}^{\tilde{m}} \frac{y_{j_i}^T s_{j_i}}{s_{j_i}^T s_{j_i}} \frac{s_{j_i}^T s_{j_i}}{s_{j_i}^T B_k^{(i-1)} s_{j_i}}$$

$$\geq \det(B_k^{(0)}) \left( \frac{\hat{\lambda}}{C_1} \right)^{\tilde{m}}$$

$$\geq C_2, \tag{A.17}$$

for some positive constant $C_2$, where the above inequalities are due to the fact that the largest eigenvalue of $B_k^{(i)}$ is less than $C_1$ and Assumption $A.2$.

The trace (A.16) and determinant (A.17) inequalities derived above imply that largest eigenvalues of all matrices $B_k$ are bounded above, uniformly, and that the smallest eigenvalues of all matrices $B_k$ are bounded away from zero, uniformly. □

## A.3   Proof of Theorem 3.2 (Strongly Convex Case)

Utilizing the result from Lemma 3.1, we now prove a linear convergence result to a neighborhood of the optimal solution, for the case where Assumptions $A$ hold.

**Theorem 3.2.** *Suppose that Assumptions A.1-A.4 above hold, and let $F^\star = F(w^\star)$, where $w^\star$ is the minimizer of $F$. Let $\{w_k\}$ be the iterates generated by the multi-batch L-BFGS method (Algorithm 1) with*

$$\alpha_k = \alpha \in (0, \frac{1}{2\mu_1 \lambda}),$$

*starting from $w_0$. Then for all $k \geq 0$,*

$$\mathbb{E}[F(w_k) - F^\star] \leq (1 - 2\alpha\mu_1\lambda)^k [F(w_0) - F^\star] + [1 - (1 - \alpha\mu_1\lambda)^k]\frac{\alpha\mu_2^2\gamma^2\Lambda}{4\mu_1\lambda}$$

$$\xrightarrow{k\to\infty} \frac{\alpha\mu_2^2\gamma^2\Lambda}{4\mu_1\lambda}.$$

*Proof.* We have that

$$F(w_{k+1}) = F(w_k - \alpha H_k \nabla F^{S_k}(w_k))$$

$$\leq F(w_k) + \nabla F(w_k)^T (-\alpha H_k \nabla F^{S_k}(w_k)) + \frac{\Lambda}{2}\|\alpha H_k \nabla F^{S_k}(w_k)\|^2$$

$$\leq F(w_k) - \alpha\nabla F(w_k)^T H_k \nabla F^{S_k}(w_k) + \frac{\alpha^2\mu_2^2\Lambda}{2}\|\nabla F^{S_k}(w_k)\|^2, \qquad (A.18)$$

where the first inequality arises due to (A.9), and the second inequality arises as a consequence of Lemma 3.1.

Taking the expectation (over $S_k$) of equation (A.18)

$$\mathbb{E}_{S_k}[F(w_{k+1})] \leq F(w_k) - \alpha\nabla F(w_k)^T H_k \nabla F(w_k) + \frac{\alpha^2\mu_2^2\Lambda}{2}\mathbb{E}_{S_k}\left[\|\nabla F^{S_k}(w_k)\|\right]^2$$

$$\leq F(w_k) - \alpha\mu_1\|\nabla F(w_k)\|^2 + \frac{\alpha^2\mu_2^2\gamma^2\Lambda}{2}, \qquad (A.19)$$

where in the first inequality we make use of Assumption $A.5$, and the second inequality arises due to Lemma 3.1 and Assumption $A.4$.

Since $F$ is $\lambda$-strongly convex, we can use the following relationship between the norm of the gradient squared, and the distance of the $k$-th iterate from the optimal solution.

$$2\lambda[F(w_k) - F^\star] \leq \|\nabla F(w_k)\|^2.$$

Using the above with (A.19),

$$\mathbb{E}_{S_k}[F(w_{k+1})] \leq F(w_k) - \alpha\mu_1\|\nabla F(w_k)\|^2 + \frac{\alpha^2\mu_2^2\gamma^2\Lambda}{2}$$

$$\leq F(w_k) - 2\alpha\mu_1\lambda[F(w_k) - F^\star] + \frac{\alpha^2\mu_2^2\gamma^2\Lambda}{2}. \qquad (A.20)$$

Let

$$\phi_k = \mathbb{E}[F(w_k) - F^\star], \qquad (A.21)$$

where the expectation is over all batches $S_0, S_1, ..., S_{k-1}$ and all history starting with $w_0$. Thus (A.20) can be expressed as,

$$\phi_{k+1} \leq (1 - 2\alpha\mu_1\lambda)\phi_k + \frac{\alpha^2\mu_2^2\gamma^2\Lambda}{2}, \tag{A.22}$$

from which we deduce that in order to reduce the value with respect to the previous function value, the step length needs to be in the range

$$\alpha \in \left(0, \frac{1}{2\mu_1\lambda}\right).$$

Subtracting $\frac{\alpha\mu_2^2\gamma^2\Lambda}{4\mu_1\lambda}$ from either side of (A.22) yields

$$\phi_{k+1} - \frac{\alpha\mu_2^2\gamma^2\Lambda}{4\mu_1\lambda} \leq (1 - 2\alpha\mu_1\lambda)\phi_k + \frac{\alpha^2\mu_2^2\gamma^2\Lambda}{2} - \frac{\alpha\mu_2^2\gamma^2\Lambda}{4\mu_1\lambda},$$

$$= (1 - 2\alpha\mu_1\lambda)\left[\phi_k - \frac{\alpha\mu_2^2\gamma^2\Lambda}{4\mu_1\lambda}\right]. \tag{A.23}$$

Recursive application of (A.23) yields

$$\phi_k - \frac{\alpha\mu_2^2\gamma^2\Lambda}{4\mu_1\lambda} \leq (1 - 2\alpha\mu_1\lambda)^k\left[\phi_0 - \frac{\alpha\mu_2^2\gamma^2\Lambda}{4\mu_1\lambda}\right],$$

and thus,

$$\phi_k \leq (1 - 2\alpha\mu_1\lambda)^k\phi_0 + \left[1 - (1 - \alpha\mu_1\lambda)^k\right]\frac{\alpha\mu_2^2\gamma^2\Lambda}{4\mu_1\lambda}. \tag{A.24}$$

Finally using the definition of $\phi_k$ (A.21) with the above expression yields the desired result,

$$\mathbb{E}\left[F(w_k) - F^\star\right] \leq \left(1 - 2\alpha\mu_1\lambda\right)^k\left[F(w_0) - F^\star\right] + \left[1 - (1 - \alpha\mu_1\lambda)^k\right]\frac{\alpha\mu_2^2\gamma^2\Lambda}{4\mu_1\lambda}. \qquad \square$$

## A.4 Proof of Lemma 3.3 (Nonconvex Case)

The following Lemma shows that the eigenvalues of the matrices generated by the multi-batch L-BFGS method are bounded above and away from zero (nonconvex case).

**Lemma 3.3.** *Suppose that Assumptions B.1-B.2 hold and let $\epsilon > 0$ be given. Let $\{H_k\}$ be the Hessian approximations generated by the multi-batch L-BFGS method (Algorithm 1), with the modification that the Hessian approximation $H_k$ update is performed only when*

$$y_k^T s_k \geq \epsilon\|s_k\|^2, \tag{A.25}$$

*else $H_{k+1} = H_k$. Then, there exist constants $0 < \mu_1 \leq \mu_2$ such that*

$$\mu_1 I \preceq H_k \preceq \mu_2 I, \qquad for \ k = 0, 1, 2, \ldots$$

*Proof.* Similar to the proof of Lemma 3.1, we study the direct Hessian approximation $B_k = H_k^{-1}$.

The curvature pairs $s_k$ and $y_k$ are updated via the following formulae

$$y_{k+1} = g_{k+1}^{O_k} - g_k^{O_k}, \qquad s_k = w_{k+1} - w_k. \tag{A.26}$$

The skipping mechanism (A.25) provides both an upper and lower bound on the quantity $\frac{\|y_k\|^2}{y_k^T s_k}$, which in turn ensures that the initial L-BFGS Hessian approximation is bounded above and away from zero. The lower bound is attained by repeated application of Cauchy's inequality to condition (A.25). We have from (A.25) that

$$\epsilon\|s_k\|^2 \leq y_k^T s_k \leq \|y_k\|\|s_k\|,$$

and therefore

$$\|s_k\| \leq \frac{1}{\epsilon}\|y_k\|.$$

It follows that

$$s_k^T y_k \leq \|s_k\|\|y_k\| \leq \frac{1}{\epsilon}\|y_k\|^2$$

and hence

$$\frac{\|y_k\|^2}{s_k^T y_k} \geq \epsilon. \tag{A.27}$$

The upper bound is attained by the Lipschitz continuity of sample gradients,

$$y_k^T s_k \geq \epsilon\|s_k\|^2$$
$$\geq \epsilon\frac{\|y_k\|^2}{\Lambda_{O_k}^2},$$

Re-arranging the above expression yields the desired upper bound,

$$\frac{\|y_k\|^2}{s_k^T y_k} \leq \frac{\Lambda_{O_k}^2}{\epsilon}. \tag{A.28}$$

Combining (A.27) and (A.28),

$$\epsilon \leq \frac{\|y_k\|^2}{y_k^T s_k} \leq \frac{\Lambda_{O_k}^2}{\epsilon}.$$

The above proves that the eigenvalues of the matrices $B_k^{(0)} = \frac{y_k^T y_k}{s_k^T y_k} I$ at the start of the L-BFGS update cycles are bounded above and away from zero, for all $k$. The rest of the proof follows the same trace-determinant argument as in the proof of Lemma 3.1, the only difference being that the last inequality in A.17 comes as a result of the cautious update strategy. □

## A.5  Proof of Theorem 3.4 (Nonconvex Case)

Utilizing the result from Lemma 3.3, we can now establish the following result about the behavior of the gradient norm for the multi-batch L-BFGS method with a cautious update strategy.

**Theorem 3.4.** *Suppose that Assumptions B.1-B.5 above hold. Let $\{w_k\}$ be the iterates generated by the multi-batch L-BFGS method (Algorithm 1) with*

$$\alpha_k = \alpha \in (0, \frac{\mu_1}{\mu_2^2 \eta \Lambda}),$$

*where $w_0$ is the starting point. Also, suppose that if*

$$y_k^T s_k < \epsilon\|s_k\|^2,$$

*for some $\epsilon > 0$, the inverse L-BFGS Hessian approximation is skipped, $H_{k+1} = H_k$. Then, for all $k \geq 0$,*

$$\mathbb{E}\Big[\frac{1}{L}\sum_{k=0}^{L-1}\|\nabla F(w_k)\|^2\Big] \leq \frac{\alpha\mu_2^2\gamma^2\Lambda}{\mu_1} + \frac{2[F(w_0) - \widehat{F}]}{\alpha\mu_1 L}$$

$$\xrightarrow{L \to \infty} \frac{\alpha\mu_2^2\gamma^2\Lambda}{\mu_1}.$$

*Proof.* Starting with (A.19),

$$\mathbb{E}_{S_k}[F(w_{k+1})] \le F(w_k) - \alpha\mu_1\|\nabla F(w_k)\|^2 + \frac{\alpha^2\mu_2^2\Lambda}{2}\mathbb{E}_{S_k}\left[\|\nabla F^{S_k}(w_k)\|\right]^2$$

$$\le F(w_k) - \alpha\mu_1\|\nabla F(w_k)\|^2 + \frac{\alpha^2\mu_2^2\Lambda}{2}(\gamma^2 + \eta\|\nabla F(w)\|^2)$$

$$= F(w_k) - \alpha\left(\mu_1 - \frac{\alpha\mu_2^2\eta\Lambda}{2}\right)\|\nabla F(w_k)\|^2 + \frac{\alpha^2\mu_2^2\gamma^2\Lambda}{2}$$

$$\le F(w_k) - \frac{\alpha\mu_1}{2}\|\nabla F(w_k)\|^2 + \frac{\alpha^2\mu_2^2\gamma^2\Lambda}{2},$$

where the second inequality holds due to Assumption $B.4$, and the fourth inequality is obtained by using the upper bound on the step length. Taking an expectation over all batches $S_0, S_1, ..., S_{k-1}$ and all history starting with $w_0$ yields

$$\phi_{k+1} - \phi_k \le -\frac{\alpha\mu_1}{2}\mathbb{E}\|\nabla F(w_k)\|^2 + \frac{\alpha^2\mu_2^2\gamma^2\Lambda}{2}, \tag{A.29}$$

where $\phi_k = \mathbb{E}[F(w_k)]$. Summing (A.29) over the first $L-1$ iterations

$$\sum_{k=0}^{L-1}[\phi_{k+1} - \phi_k] \le -\frac{\alpha\mu_1}{2}\sum_{k=0}^{L-1}\mathbb{E}\|\nabla F(w_k)\|^2 + \sum_{k=0}^{L-1}\frac{\alpha^2\mu_2^2\gamma^2\Lambda}{2}$$

$$= -\frac{\alpha\mu_1}{2}\mathbb{E}\left[\sum_{k=0}^{L-1}\|\nabla F(w_k)\|^2\right] + \frac{\alpha^2\mu_2^2\gamma^2\Lambda L}{2}. \tag{A.30}$$

The left-hand-side of the above inequality is a telescoping sum

$$\sum_{k=0}^{L-1}[\phi_{k+1} - \phi_k] = \phi_L - \phi_0$$

$$= \mathbb{E}[F(w_L)] - F(w_0)$$

$$\ge \widehat{F} - F(w_0).$$

Substituting the above expression into (A.30) and re-arranging terms

$$\mathbb{E}\left[\sum_{k=0}^{L-1}\|\nabla F(w_k)\|^2\right] \le \frac{\alpha\mu_2^2\gamma^2\Lambda L}{\mu_1} + \frac{2[F(w_0) - \widehat{F}]}{\alpha\mu_1}.$$

Dividing the above equation by $L$ completes the proof. $\qquad\square$

# B Extended Numerical Experiments - Real Datasets

In this Section, we present further numerical results, on the datasets listed in Table 1, in both the multi-batch and fault-tolerant settings. Note, that some of the datasets are too small, and thus, there is no reason to run them on a distributed platform; however, we include them as they are part of the standard benchmarking datasets.

**Notation.** Let $n$ denote the number of training samples in a given dataset, $d$ the dimension of the parameter vector $w$, and $K$ the number of MPI processes used. The parameter $r$ denotes the fraction of samples in the dataset used to define the gradient, i.e., $r = \frac{|S|}{n}$. The parameter $o$ denotes the length of overlap between consecutive samples, and is defined as a fraction of the number of samples in a given batch $S$, i.e., $o = \frac{|O|}{|S|}$.

Table 1: Datasets together with basic statistics. All datasets are available at `https://www.csie.ntu.edu.tw/~cjlin/libsvmtools/datasets/binary.html`.

| Dataset | $n$ | $d$ | Size (MB) | K |
|---|---|---|---|---|
| ijcnn (test) | 91,701 | 22 | 14 | 4 |
| cov | 581,012 | 54 | 68 | 4 |
| news20 | 19,996 | 1,355,191 | 134 | 4 |
| rcvtest | 677,399 | 47,236 | 1,152 | 16 |
| url | 2,396,130 | 3,231,961 | 2,108 | 16 |
| kdda | 8,407,752 | 20,216,830 | 2,546 | 16 |
| kddb | 19,264,097 | 29,890,095 | 4,894 | 16 |
| webspam | 350,000 | 16,609,143 | 23,866 | 16 |
| splice-site | 50,000,000 | 11,725,480 | 260,705 | 16 |

We focus on logistic regression classification; the objective function is given by

$$\min_{w \in \mathbb{R}^d} F(w) = \frac{1}{n} \sum_{i=1}^{n} \log(1 + e^{-y^i(w^T x^i)}) + \frac{\sigma}{2} \|w\|^2,$$

where $(x^i, y^i)_{i=1}^n$ denote the training examples and $\sigma = \frac{1}{n}$ is the regularization parameter.

## B.1 Multi-batch L-BFGS Implementation

For the experiments in this section (Figures 5-13), we run four methods:

- (Robust L-BFGS) robust multi-batch L-BFGS (Algorithm 1),
- (L-BFGS) multi-batch L-BFGS without enforcing sample consistency; gradient differences are computed using different samples, i.e., $y_k = g_{k+1}^{S_{k+1}} - g_k^{S_k}$,
- (Gradient Descent) multi-batch gradient descent; obtained by setting $H_k = I$ in Algorithm 1,
- (SGD) serial SGD; at every iteration one sample is used to compute the gradient.

In Figures 5-13 we show the evolution of $\|\nabla F(w)\|$ for different step lengths $\alpha$, and for various batch ($|S| = r \cdot n$) and overlap ($|O| = o \cdot |S|$) sizes. Each Figure (5-13) consists of 10 plots that illustrate the performance of the methods with the following parameters:

- Top 3 plots: $\alpha = 1$, $o = 20\%$ and $r = 1\%, 5\%, 10\%$
- Middle 3 plots: $\alpha = 0.1$, $o = 20\%$ and $r = 1\%, 5\%, 10\%$
- Bottom 4 plots: $\alpha = 1$, $r = 1\%$ and $o = 5\%, 10\%, 20\%, 30\%$

As is expected for quasi-Newton methods, robust L-BFGS performs best with a step-size $\alpha = 1$, for the most part.

Figure 5: **ijcnn1 dataset**. Comparison of Robust L-BFGS, L-BFGS (multi-batch L-BFGS without enforcing sample consistency), Gradient Descent (multi-batch Gradient method) and SGD. Top part: we used $\alpha \in \{1, 0.1\}$, $r \in \{1\%, 5\%, 10\%\}$ and $o = 20\%$. Bottom part: we used $\alpha = 1$, $r = 1\%$ and $o \in \{5\%, 10\%, 20\%, 30\%\}$. Solid lines show average performance, and dashed lines show worst and best performance, over 10 runs (per algorithm). $K = 4$ MPI processes.

Figure 6: **cov dataset**. Comparison of Robust L-BFGS, L-BFGS (multi-batch L-BFGS without enforcing sample consistency), Gradient Descent (multi-batch Gradient method) and SGD. Top part: we used $\alpha \in \{1, 0.1\}$, $r \in \{1\%, 5\%, 10\%\}$ and $o = 20\%$. Bottom part: we used $\alpha = 1$, $r = 1\%$ and $o \in \{5\%, 10\%, 20\%, 30\%\}$. Solid lines show average performance, and dashed lines show worst and best performance, over 10 runs (per algorithm). $K = 4$ MPI processes.

Figure 7: **news20 dataset**. Comparison of Robust L-BFGS, L-BFGS (multi-batch L-BFGS without enforcing sample consistency), Gradient Descent (multi-batch Gradient method) and SGD. Top part: we used $\alpha \in \{1, 0.1\}$, $r \in \{1\%, 5\%, 10\%\}$ and $o = 20\%$. Bottom part: we used $\alpha = 1$, $r = 1\%$ and $o \in \{5\%, 10\%, 20\%, 30\%\}$. Solid lines show average performance, and dashed lines show worst and best performance, over 10 runs (per algorithm). $K = 4$ MPI processes.

Figure 8: **rcvtest dataset**. Comparison of Robust L-BFGS, L-BFGS (multi-batch L-BFGS without enforcing sample consistency), Gradient Descent (multi-batch Gradient method) and SGD. Top part: we used $\alpha \in \{1, 0.1\}$, $r \in \{1\%, 5\%, 10\%\}$ and $o = 20\%$. Bottom part: we used $\alpha = 1$, $r = 1\%$ and $o \in \{5\%, 10\%, 20\%, 30\%\}$. Solid lines show average performance, and dashed lines show worst and best performance, over 10 runs (per algorithm). $K = 16$ MPI processes.

Figure 9: **url dataset**. Comparison of Robust L-BFGS, L-BFGS (multi-batch L-BFGS without enforcing sample consistency), Gradient Descent (multi-batch Gradient method) and SGD. Top part: we used $\alpha \in \{1, 0.1\}$, $r \in \{1\%, 5\%, 10\%\}$ and $o = 20\%$. Bottom part: we used $\alpha = 1$, $r = 1\%$ and $o \in \{5\%, 10\%, 20\%, 30\%\}$. Solid lines show average performance, and dashed lines show worst and best performance, over 10 runs (per algorithm). $K = 16$ MPI processes.

Figure 10: **kdda dataset**. Comparison of Robust L-BFGS, L-BFGS (multi-batch L-BFGS without enforcing sample consistency), Gradient Descent (multi-batch Gradient method) and SGD. Top part: we used $\alpha \in \{1, 0.1\}$, $r \in \{1\%, 5\%, 10\%\}$ and $o = 20\%$. Bottom part: we used $\alpha = 1$, $r = 1\%$ and $o \in \{5\%, 10\%, 20\%, 30\%\}$. Solid lines show average performance, and dashed lines show worst and best performance, over 10 runs (per algorithm). $K = 16$ MPI processes.

Figure 11: **kddb dataset**. Comparison of Robust L-BFGS, L-BFGS (multi-batch L-BFGS without enforcing sample consistency), Gradient Descent (multi-batch Gradient method) and SGD. Top part: we used $\alpha \in \{1, 0.1\}$, $r \in \{1\%, 5\%, 10\%\}$ and $o = 20\%$. Bottom part: we used $\alpha = 1$, $r = 1\%$ and $o \in \{5\%, 10\%, 20\%, 30\%\}$. Solid lines show average performance, and dashed lines show worst and best performance, over 10 runs (per algorithm). $K = 16$ MPI processes.

Figure 12: **webspam dataset**. Comparison of Robust L-BFGS, L-BFGS (multi-batch L-BFGS without enforcing sample consistency), Gradient Descent (multi-batch Gradient method) and SGD. Top part: we used $\alpha \in \{1, 0.1\}$, $r \in \{1\%, 5\%, 10\%\}$ and $o = 20\%$. Bottom part: we used $\alpha = 1$, $r = 1\%$ and $o \in \{5\%, 10\%, 20\%, 30\%\}$. Solid lines show average performance, and dashed lines show worst and best performance, over 10 runs (per algorithm). $K = 16$ MPI processes.

Figure 13: **splice-cite dataset**. Comparison of Robust L-BFGS, L-BFGS (multi-batch L-BFGS without enforcing sample consistency), Gradient Descent (multi-batch Gradient method) and SGD. Top part: we used $\alpha \in \{1, 0.1\}$, $r \in \{1\%, 5\%, 10\%\}$ and $o = 20\%$. Bottom part: we used $\alpha = 1$, $r = 1\%$ and $o \in \{5\%, 10\%, 20\%, 30\%\}$. Solid lines show average performance, and dashed lines show worst and best performance, over 10 runs (per algorithm). $K = 16$ MPI processes. (No Serial SGD experiments due to memory limitations of our cluster.)

## B.2  Fault-tolerant L-BFGS Implementation

If we run a distributed algorithm, for example on a shared computer cluster, then we may experience delays. Such delays can be caused by other processes running on the same compute node, node failures and for other reasons. As a result, given a computational (time) budget, these delays may cause nodes to fail to return a value. To illustrate this behavior, and to motivate the robust fault-tolerant L-BFGS method, we run a simple benchmark MPI code on two different environments:

- **Amazon EC2** – Amazon EC2 is a cloud system provided by Amazon. It is expected that if load balancing is done properly, the execution time will have small noise; however, the network and communication can still be an issue. (4 MPI processes)

- **Shared Cluster** – In our shared cluster, multiple jobs run on each node, with some jobs being more demanding than others. Even though each node has 16 cores, the amount of resources each job can utilize changes over time. In terms of communication, we have a GigaBit network. (11 MPI processes, running on 11 nodes)

We run a simple code on the cloud/cluster, with MPI communication. We generate two matrices $A, B \in R^{n \times n}$, then synchronize all MPI processes and compute $C = A \cdot B$ using the GSL C-BLAS library. The time is measured and recorded as computational time. After the matrix product is computed, the result is sent to the master/root node using asynchronous communication, and the time required is recorded. The process is repeated 3000 times.

Figure 14: Distribution of Computation and Communication Time for Amazon EC2 and Shared Cluster. Figures show worst and best time, average time and 10% and 90% quantiles. Amazon Cloud EC: In the experiment: 4 MPI processes; Shared Cluster: 11 MPI processes.

The results of the experiment described above are captured in Figure 14. As expected, on the Amazon EC2 cloud, the matrix-matrix multiplication takes roughly the same time for all replications and the noise in communication is relatively small. In this example the cost of communication is negligible when compared to the cost of computation. On our shared cluster, one cannot guarantee that all resources are exclusively used for a specific process, and thus, the computation and communication time is considerably more stochastic and unbalanced. For some cases the difference between the minimum and maximum computation (communication) time varies by an order of magnitude or more. Hence, on such a platform a fault-tolerant algorithm that only uses information from nodes that return an update within a preallocated budget is a natural choice.

In Figures 15-19 we show a comparison of the proposed robust multi-batch L-BFGS method and the multi-batch L-BFGS method that does not enforce sample consistency (L-BFGS). In these experiments, $p$ denotes the probability that a single node (MPI process) will not return a gradient evaluated on local data within a given time budget. We illustrate the performance of the methods for $\alpha = 0.1$ and $p \in \{0.1, 0.2, 0.3, 0.4, 0.5\}$. We observe that the robust implementation is not affected much by the failure probability $p$.

Figure 15: **rcvtest dataset**. Comparison of Robust L-BFGS and L-BFGS in the presence of faults. We used $\alpha = 0.1$ and $p \in \{0.1, 0.2, 0.3, 0.4, 0.5\}$. Solid lines show average performance, and dashed lines show worst and best performance, over 10 runs (per algorithm). $K = 16$ MPI processes.

Figure 16: **webspam dataset**. Comparison of Robust L-BFGS and L-BFGS in the presence of faults. We used $\alpha = 0.1$ and $p \in \{0.1, 0.2, 0.3, 0.4, 0.5\}$. Solid lines show average performance, and dashed lines show worst and best performance, over 10 runs (per algorithm). $K = 16$ MPI processes.

Figure 17: **kdda dataset**. Comparison of Robust L-BFGS and L-BFGS in the presence of faults. We used $\alpha = 0.1$ and $p \in \{0.1, 0.2, 0.3, 0.4, 0.5\}$. Solid lines show average performance, and dashed lines show worst and best performance, over 10 runs (per algorithm). $K = 16$ MPI processes.

Figure 18: **kddb dataset**. Comparison of Robust L-BFGS and L-BFGS in the presence of faults. We used $\alpha = 0.1$ and $p \in \{0.1, 0.2, 0.3, 0.4, 0.5\}$. Solid lines show average performance, and dashed lines show worst and best performance, over 10 runs (per algorithm). $K = 16$ MPI processes.

Figure 19: **url dataset**. Comparison of Robust L-BFGS and L-BFGS in the presence of faults. We used $\alpha = 0.1$ and $p \in \{0.1, 0.2, 0.3, 0.4, 0.5\}$. Solid lines show average performance, and dashed lines show worst and best performance, over 10 runs (per algorithm). $K = 16$ MPI processes.

# C  Scaling of Robust Multi-Batch L-BFGS Implementation

In this Section, we study the strong and weak scaling properties of the robust multi-batch L-BFGS method on an artificial dataset. For various values of $r$ and $K$, we measure the time needed to compute a gradient (Gradient) and the time needed to compute and communicate the gradient (Gradient+C), as well as, the time needed to compute the L-BFGS direction (L-BFGS) and the associated communication overhead (L-BFGS+C).

## C.1  Strong Scaling

Figure 20 depicts the strong scaling properties of our proposed algorithm. We generate a dataset with $n = 10^7$ samples and $d = 10^4$ dimensions, where each sample has 160 randomly chosen non-zero elements (dataset size 24GB). We run our code for different values of $r$ (different batch sizes $S_k$), with $K = 1, 2, \ldots, 128$ number of MPI processes.

One can observe that the compute time for the gradient and the L-BFGS direction decreases as $K$ is increased. However, when communication time is considered, the combined cost increases slightly as $K$ is increased. Notice that for large $K$, even when $r = 10\%$ (i.e., $10\%$ of all samples processed in one iteration, $\sim$18MB of data), the amount of local work is not sufficient to overcome the communication cost.

Figure 20: Strong scaling of robust multi-batch L-BFGS on a problem with artificial data; $n = 10^7$ and $d = 10^4$. Each sample has 160 non-zero elements. $+C$ indicates that we include communication time to the gradient computation and L-BFGS update computation.

## C.2  Weak Scaling - Fixed Problem Dimension, Increasing Data Size

In order to illustrate the weak scaling properties of the algorithm, we generate a data-matrix $X \in R^{10^7 \times 10^4}$, and run it on a shared cluster with $K = 1, 2, 4, 8, \ldots, 128$ MPI processes. For a given number of MPI processes $(K)$, each sample contains $10 \cdot K$ non-zero elements. Effectively, the dimension of the problem is fixed, but sparsity of the data is decreased as more MPI processes are used. The size of the input data is $1.5 \cdot K$ GB (i.e., 1.5GB per MPI process).

The compute time for the gradient is almost constant, this is because the amount of work per MPI process (rank) is almost identical; see Figure 21. On the other hand, because we are using a Vector-Free L-BFGS implementation [9] for computing the L-BFGS direction, the amount of time needed for each node to compute the L-BFGS direction is decreasing as $K$ is increased. However, increasing $K$ does lead to larger communication overhead, which can be observed in Figure 21. For $K = 128$ (192GB of data) and $r = 10\%$, almost 20GB of data are processed per iteration in less than 0.1 seconds, which implies that one epoch would take around 1 second.

Figure 21: Weak scaling of robust multi-batch L-BFGS on a problem with artificial data; $n = 10^7$ and $d = 10^4$. Each sample has $10 \cdot K$ non-zero elements. $+C$ indicates that we also include communication time to the gradient computation and L-BFGS update computation.

## C.3  Increasing Problem Dimension, Fixed Data Size and $K$

In this experiment, we investigate the effect of a change in the dimension $d$ of the problem on the performance of the algorithm. We fix the size of data ($29GB$) and the number of MPI processes ($K = 8$). We generate data with $n = 10^7$ samples, where each sample has 200 non-zero elements. Figure 22 shows that increasing the dimension $d$ has a mild effect on the computation time of the gradient, while the effect on the time needed to compute the L-BFGS direction is more apparent. However, if communication time is taken into consideration, the time required for the gradient computation and the L-BFGS direction computation increase as $d$ is increased.

Figure 22: Scaling of robust multi-batch L-BFGS on a problem with artificial data; $n = 10^7$ samples, with increasing $d$ and $K = 8$ MPI processes. Each sample had 200 non-zero elements. $+C$ indicates that we also include communication time to the gradient computation and L-BFGS update computation.