[Reviews · NeurIPS 2016]

Reviewer 1

Summary

In supervised learning, one is interested in minimizing the empirical risk where efficient optimization algorithms become the key. First-order methods such as stochastic gradient descent and its variants are reasonably well understood admitting efficient implementation and parallelization techniques. However there has been a recent interest in making second-order methods such as Newton's method or L-BFGS method efficient for such large-scale problems. This paper is along this direction, presenting a new variant of the stochastic L-BFGS method that is efficient and robust in mainly two settings: The first arises in the presence of node failures in a distributed computing environment, the second occurs when one uses an adaptive batch size that varies over iterations for accelerating learning. The main idea is to form the Hessian estimate based on the overlap between consecutive batches (the intuition why this works is that we have less limitation in choosing the second-order information matrix compared to an estimate of the true gradient). This way, the Hessian corrections and updates with L-BFGS become more stable. The results are also illustrated and supported by numerical experiments. I would recommend this paper to be accepted. My minor concerns/questions are below: 1) If the strong convexity lower-bound mu_1 is very small, then it seems that the resulting limit in Theorem 3.2 will be worse than the limit obtained by a first-order version of your algorithm when one would choose simply H_k = I. Could you please add a few lines to clarify this point? 2) I would suggest to add a few more references to the intro regarding recent work on stochastic L-BFGS for the sake of completeness of the literature survey. Examples include [Moritz, Nishihara, Jordan, 2016], [Mokhtari, Ribeiro, 2014] 3) To my knowledge, [Schmidt, Roux, Bach] paper appeared in Math. Programming recently. I would suggest to give a reference to the Math. Programming journal instead of the reference to the arxiv version. 4) Typo in line 59: use --> used

Qualitative Assessment

The paper is readable and well-written on an interesting topic that has received much recent attention.

Confidence in this Review

2-Confident (read it all; understood it all reasonably well)


Reviewer 2

Summary

The paper studies a multi-batch L-BFGS method that is robust in the absence of sample consistency. Their experiments show its effectiveness on logistic regression problems.

Qualitative Assessment

The proposed robust Quasi-Newton method is interesting and new with regard to using overlapping consecutive samples. The methods works both in node failures or multi-batch settings. The paper suggests two multi-batch sampling strategies, also analyzes the convergence properties of the multi-batch L-BFGS method. The numerical results show the proposed method is more efficient than comparing algorithms including SGD for logistic problems. The paper is clearly written.

Confidence in this Review

2-Confident (read it all; understood it all reasonably well)


Reviewer 3

Summary

In this paper, the author considered the a multi-batch L-BFGS method for optimizing logistic regression problem. In stand stochastic L-BFGS, the algorithm can be instable due to the calculation of gradient difference when the batch changes. The authors present a robust multi-batch L-BFGS, in which each computation node contain a subset of the whole data set with some overlapping elements. In this way, a robust quasi-Newton updating can be proposed. Two sample strategies were proposed to constructing subsets with overlapping elements. A convergence analysis regarding strongly convex objective functions was conducted. Extensive experiments (included in the supplementary file) demonstrate the superiority of the proposed method over some baselines.

Qualitative Assessment

This paper is easy to follow and interesting. My major concerns are on the experiments. First, the comparison to SGD in the paper is not very fair. A mini-batch SGD with parallel implementation is required for the comparison. Recently, there are many improved mini-bath (dual-primal) SGD methods, e.g. those attached below. Otherwise, I am not sure the superiority over the SGD methods. Lee J, Ma T, Lin Q. Distributed stochastic variance reduced gradient methods[J]. arXiv preprint arXiv:1507.07595, 2015. Shalev-Shwartz S, Zhang T. Accelerated mini-batch stochastic dual coordinate ascent[C]//Advances in Neural Information Processing Systems. 2013: 378-385. Since the comparison is on classification tasks, a comparison of prediction accuracy w.r.t. running time (not epochs) is required. The paper analyzed the nonconvex cases, but no experiment is performed to verify the theoretical results. Another question is about convergence speed on convex cases (but not strongly convex). For example, the convergence on the L1-norm regularized SVM problem. Line 164: randomly selecting elements from S_k?

Confidence in this Review

3-Expert (read the paper in detail, know the area, quite certain of my opinion)


Reviewer 4

Summary

This paper proposes a multi-batch L-BFGS method to obtain fast learning processes. One main issue for the method is instability, led by the sample inconsistency between the batches for two consecutive iterations. The authors proposed overlapping batches. Then stable second-order information is captured by computing the gradient difference for the overlapping subset of samples. This paper also provides theoretical guarantees, where given strongly convex f and reasonable assumptions on the batches, the iterates linearly converge to a neighborhood of the optimum. For non-convex f, the average norm of the gradient over iterations decreases as a constant plus O(1/t) term. (not converging to zero). The numerical results show that the convergence of the algorithm over epochs is competitive against SGD. It is also shown that the proposed algorithm is much more stable than just using multi-batch L-BFGS without batch overlapping.

Qualitative Assessment

This paper provides an interesting attempt to give competitive practical performance of second-order methods against first-order ones in the distributed and stochastic setting. The convergence analysis is neat but not surprising. As stated in the beginning, when the communication cost is limited, the proposed algorithm is more likely to outperform SGD. It would have been impressive if the authors conducted experiments with such settings.

Confidence in this Review

2-Confident (read it all; understood it all reasonably well)


Reviewer 5

Summary

This paper proposes a variant of L-BFGS that can be used in a distributed computing environment.

Qualitative Assessment

This paper is well-written and effectively conveys the core idea. Unlike the first order method, subsampling of "inverse Hessian matrix" in the second order method is too noisy, and problematic, which makes a second-order method hard to be applied in distributed computing environment. This paper proposes to handle "noiseness" of the inverse Hessian matrix by using overlapped version of estimates, instead of using non-overlapped version of estimates. I found this idea to be interesting, and preliminary results look promising.

Confidence in this Review

2-Confident (read it all; understood it all reasonably well)


Reviewer 6

Summary

The article tackles an original approach to ease the parallelism from the batch method L-BFGS. The authors explain the heuristics behind the algorithm, provide a convergence analysis in both the convex and the nonconvex settings, and show the performance of their method on a real dataset.

Qualitative Assessment

I liked the article which presented clear ideas and focus on a new side of stochastic descent algorithm. The article aims at designing stochastic version of L-BFGS. At each iteration, you sample a large set of index (while SGD-based algorithms with mini-batchs use small mini-batchs), and select some indices to update the estimate of the inverse Hessian matrix (just like the original L-BFGS) The goal of the paper is to show that using quasi-Newton updates with overlap between consecutive batches ensure stability. I liked the convergence analysis: even though the optimality gap (in the convex case) doesn't converge to 0, the speed of convergence if fast. I would have appreciated more comments about the bounds (especially the denominator of the bound in Theorem 4),.

Confidence in this Review

2-Confident (read it all; understood it all reasonably well)